# Health Benefits of Antioxidant Bioactive Compounds in Ginger (*Zingiber officinale*) Leaves by Network Pharmacology Analysis Combined with Experimental Validation

**DOI:** 10.3390/antiox13060652

**Published:** 2024-05-27

**Authors:** Dong-Geon Nam, Mina Kim, Ae-Jin Choi, Jeong-Sook Choe

**Affiliations:** Division of Functional Food & Nutrition, Department of Agrofood Resources, National Institute of Agricultural Sciences, Rural Development Administration, Wanju-gun 55365, Republic of Korea; developfoods@gmail.com (D.-G.N.); lucidminakim@gmail.com (M.K.); aejini77@korea.kr (A.-J.C.)

**Keywords:** ginger leaves, chemical diversity, flavonoid, network pharmacology, molecular docking

## Abstract

Network pharmacology is an ideal tool to explore the effects of therapeutic components derived from plants on human metabolic diseases that are linked to inflammation. This study investigated the antioxidant effects of ginger leaves (GLs) and predicted targets for antioxidant activity. Quantitative and free radical scavenging analyses were performed to detect the main bioactive compounds of GLs and evaluate their antioxidant activities. Chemical diversity and network pharmacology approaches were used to predict key antioxidant components of GLs and their molecular targets. Nine major bioactive compounds of GLs were quantified using an internal standard method, and the antioxidant activity was evaluated using the DPPH and ABTS free radical scavenging methods. We first built the compound-gene-pathways and protein-protein interaction networks of GLs-related antioxidant targets and then conducted gene ontology and Kyoto Encyclopedia of Gene and Genome (KEGG) pathway enrichment analyses. Molecular docking results show that astragalin, a compound isolated from GLs, had the highest level of connectivity in the compound-target network and was involved in inflammation-related biosynthesis by directly impacting cytokine gene expression and PTGS2 inhibition markers. These findings not only suggest that the compounds isolated from GLs can be developed as potential antioxidants, but also demonstrate the applicability of network pharmacology to assess the potential of foods for disease treatment.

## 1. Introduction

Human metabolic diseases tend to involve both genetic and environmental risk factors [1]. As humanity has gradually developed from an agricultural society to an industrial society and now to a digital society, the prevalence of metabolic diseases such as obesity, diabetes, and fatty liver disease has increased [2]. Radical lifestyle changes resulting from social development are seriously threatening human health and quality of life [3,4,5]. In an attempt to address some of these threats, phytochemicals in plants have been identified to have various health benefits, including anti-inflammatory, antioxidant, and anticancer effects [6,7].

The leaves of ginger plants are typically used compost or discarded. Recently, GLs were added to the 2016 list of “raw materials that can be used in food” by Korea’s Rural Development Administration. Traditionally, ginger has been used by many cultures for a variety of purposes, including promoting digestion, preventing motion sickness, and reducing inflammation. Several studies of the physiological activities of functional compounds found in GLs have been conducted, but their utility is limited. Clinical studies have shown ginger consumption to be an effective treatment for various conditions, including reductions in vomiting and nausea [8] and in non-acute and acute musculoskeletal pain and discomfort [9,10], and to have positive effects on inflammatory response and immune function in rheumatoid arthritis [11].

Data regarding the metabolic relevance, pharmacology, and pharmacokinetics of phytochemicals are lacking. Phytochemical actions in the human body are difficult to interpret using chemical and dose-response experiments alone [12,13,14,15] given the complex chemical composition of phytochemicals, their diverse bioactive effects, and difficulties with the standardization of preparations and dosage determination, among other factors [16,17]. Practical human applications of phytochemicals are limited by low bioavailability, differences in individual responses, lack of evidence of long-term effectiveness and safety, and interactions with other compounds [18,19,20]. To address these limitations, it is important to determine the absorption rate and metabolic pathways of such compounds and to find a consistent specific pattern.

Network pharmacology is an innovative approach to analyze mechanisms of action that involve complex interactions among multiple components, targets, and pathways [21,22]. This approach can provide an integrative perspective to explore the overall relationships between active ingredients and diseases and predict the mechanisms of action of active ingredients and the feasibility of their use as therapies [23]. Molecular docking and dynamics simulations can predict the binding mode and affinity of receptor-ligand complexes and can estimate the structural stability and dynamics of receptor-ligand interactions [24]. Recently, the application of these technologies has made it possible to determine the direction of efficacy exploration in other related fields such as herbal medicine, food, and cosmetics and to study the metabolic interactions of phytochemicals in the human body. Network pharmacology is more efficient, economical, and environmentally friendly than traditional methods used to evaluate the efficacy and safety of phytochemicals.

In this study, we combined chemical diversity, network pharmacology, and molecular docking simulations to systematically explore the biological pathways associated with the antioxidant activity of GLs.

## 2. Materials and Methods

### 2.1. Plant Materials

Ginger (*Zingiber officinale*) leaves (GLs), i.e., mature whole plants, were collected from Seosan, Chungcheongnam-do of Republic of Korea (33°22′30.5″ N 126°12′57.0″ E, 2022 years). Each sample was washed, cut into pieces, and hot-air-dried at 60 °C for 12 h (Doo Sung, DS-240BC, DS24900111, Gwangju, Republic of Korea). The dried sample was ground in a grinder (Shinil, SMX-C4000WK, Incheon, Republic of Korea), sieved to 30 mesh, stored frozen in a −70 °C cryogenic freezer (GMS, GS28130416-141, Yangju, Republic of Korea), and used as a sample. The samples were subjected to hot-water extraction (ratio 1:10 *w*/*v*, 60 min, and 90 ± 2 °C) in a constant-temperature water bath.

### 2.2. Proximate Components and Minerals

Proximate components and minerals of GLs were analyzed according to the AOAC method [25]. The moisture content was quantified by drying at 105 °C using an atmospheric pressure drying method, and the crude protein was determined using an automatic protein analyzer (Kjeltec 2400 AUT, Foss Tecator, Hilleroed, Denmark). Crude fat was extracted and quantified with diethyl ether using a Soxhlet extractor (Soxtec System HT 1043 extraction unit, Foss Tecator, Hilleroed, Denmark), and ash content was measured using the direct incineration method at 600 °C. For mineral analysis, macro minerals such as calcium and potassium and trace minerals such as iron and manganese were analyzed using a wet decomposition method. An amount of 7 mL of nitric acid solution and 1 mL of hydrogen peroxide were injected into 0.5 g of frozen samples and heated on a heating block at 200 °C for 30 min. After cooling to room temperature for 30 min to remove the acid, the decomposed sample was measured in 50 mL and measured using ICP-OES (Vista-PRO, Agilent Technologies, Waldbronn, Germany).

### 2.3. Amino Acids

Amino acid analysis referred to the method of [26]. The standards and GLs extracts were also analyzed on a Thermo HPLC system (Thermo Scientific, Karlsruhe, Germany) with operating parameters of injection volume 0.5 µL; flow rate 1.5 mL/min; retention time 35 min; wavelength 338 nm; and eluents 40 mM sodium phosphate, pH 7 (A) and ice-cold lysis/extraction buffer (methanol/acetonitrile/water, 4.5/4.5/1 *v*/*v*) (B). The gradient elution had the following profile: 0–3.0 min, 95% A; 3.0–24.0 min, 45% A; 24.0–31.0 min, 10% A; 31.0–35.0 min, 95% A. In the HPLC analysis, an Inno C18 column (5 µm, 150 × 4.6 mm, Youngjin Biochrom, Seongnam, Republic of Korea) was used and the column temperature was maintained at 40 °C. A reference amino acid spectrum was obtained by titrating a mixture of amino acids of known concentrations.

### 2.4. Identification of Phytochemicals Using UPLC-DAD-QTOF/MS

Individual flavonoid component analysis was performed using the Waters ACQUITY UPLC TM system (Waters, Milford, MA, USA) and Xevo G2-S QTOF ESI/MS (Waters MS Technologies, Manchester, UK) [27]. The analytical column was a C18 column (CORTECS^®^ UPLC^®^ T3 1.6 μm, 2.1 × 150 mm) and a pre-column (CORTECS^®^ UPLC^®^ T3 1.6 μm, VanGuardTM Pre-column 2.1 × 5 mm), and the column oven temperature was 30 °C, the sample injection volume was set to 1 μL, and the detection wavelength was set to 210–400 nm. Mobile-phase solvent A was distilled water containing 0.5% formic acid, and solvent B was acetonitrile containing 0.5% formic acid, and the flow rate was 0.3 mL/min. Qualitative analysis of flavonoids using mass spectrometry was performed in positive-ion mode. The capillary voltage was set to 3500 V, the sampling cone voltage was set to 40 V, and the extraction cone voltage was set to 4.0 V. The ion source temperature was set at 120 °C and the desolvation temperature was set at 500 °C. The desolvation gas was set at 1050 L/h, the cone gas was set at 50 L/h, and the scan range was set at *m*/*z* 200–1200. Mass spectrometry data processing, including *m*/*z*, retention time, and ion intensity, was performed using UNIFI software, version 1.8.2.169 (Waters). Material identification was conducted through online databases connected to UNIFI software. The content of individual flavonoid components was relatively quantified (%) by comparing the area of each component with the area of the internal standard (6-methoxyluteolin, λ (max) 350 nm) in a 1:1 ratio.

### 2.5. Antioxidant Properties and Radical Scavenging Activity

Total polyphenol (TP) content was measured using the F-C method, as reported by [28]. TP was calculated using the following:(1)TP(mgGAE/g)=Gallic acid×Total extract/Sample

Total flavonoid content (TF) was measured by a modified method of [29]. The total flavonoid content was obtained from a standard curve prepared by quantifying
(2)TF (mg QE/g)=(+)−Catechin×Total extract/Sample

Radical scavenging activities (RSA) were evaluated using the DPPH and ABTS assays [30]. Briefly, 0.15 mM DPPH solution was prepared in ethanol, and 0.16 mL of this solution was added to 0.04 mL of extract. The mixture was kept at room temperature for 30 min in the dark and the absorbance was recorded at 517 nm (Infinite M200 Pro, Tecan, Männedorf, Switzerland). For the production of ABTS, 7.4 mM ABTS was reacted with 2.6 mM potassium persulfate and stored in the dark at room temperature for 12 h. Then, 0.015 mL of extracts were placed in a 96-well plate, 0.285 mL of the ABTS solution was added, and the absorbance was measured at 734 nm.
(3)%RSA=[(Abs.of control−Abs.of sample)/Abs.of control]×100

### 2.6. Antioxidant-Related Target Screening

The SwissTargetPrediction tool (http://www.swisstargetprediction.ch/ accessed on 11 March 2024) was used to screen the target corresponding to the component and screen the targets based on the possible criteria before (probability > 0%) merging the targets and removing repeated values [31]. To find relevant targets, we used the search term ‘antioxidants’ to gather and combine targets from the Human Gene Database (https://www.genecards.org/ accessed on 19 March 2024) [32]. Then, the target dataset was imported into the tool called Venny 2.1.0 (https://bioinfogp.cnb.csic.es/tools/venny/index.html/ accessed on 20 March 2024) to create a Venn diagram.

### 2.7. Protein-Protein Interactions (PPI)

To build a PPI network, the common targets were imported into the STRING 12.0 database (https://www.string-db.org/ accessed on 21 March 2024). Subsequently, the acquired data were transferred into Cytoscape 3.10.1. [33,34] for topological analysis to screen out the primary targets of pathways associated with antioxidants.

### 2.8. Gene Ontology (GO) and Kyoto Encyclopedia of Genes and Genomes (KEGG) Pathway Enrichment

GO function and KEGG pathway enrichment analyses were carried out using DAVID (https://david.ncifcrf.gov/ accessed on 24 March 2024) using Homo sapiens as the selected species [35]. Functional annotation of GO involves the attribution of functional terms associated with genes in the database, relying on sequence similarity, experimental evidence, or selected studies. KEGG pathways provide a broad view of interactions between genes, proteins, and small molecules across a variety of biological processes. The R language package ‘ggplot2’ was used to visualize the GO function and KEGG pathway enrichment. The threshold point was set at *p* < 0.05 for all GO enrichment and pathway studies. The genes involved in the enriched pathway analysis were extracted along with their corresponding GO entries and KEGG relationships. The final pathway map was created by integrating and plotting the top-ranked paths.

### 2.9. Validation of Prognostic Power of Key Genes

GEPIA2 (http://gepia.cancer-pku.cn/ accessed on 1 April 2024) web server was utilized for the mRNA expression in several carcinoma cells and normal tissue.

### 2.10. Molecular Docking

In this study, bioactive compounds identified in GLs by LC/MS analysis were selected for docking studies. Chemical structures of selected bioactive compounds were retrieved through the PubChem compound database. Before proceeding with docking assays, human tumor necrosis factor (TNF, PDB ID: 2AZ5), human prostaglandin-endoperoxide synthase-2 (PTGS2, PDB ID: 5F19), and human interleukin-2 (IL2, PDB ID: 2ERJ) proteins were purified and energy-optimized. This study was converted into a protein model using UCSF Chimera software (version 1.16). UCSF Chimera and AutoDock-Vina software (version 1.2.5) were used to dock the antioxidant target proteins TNF, PTGS2, and IL2 with selected functional compounds and calculate binding energies. The binding energy and binding contacts of each ligand were obtained, and the docked complex was analyzed using PyMol (3D) and Discovery Studio Visualizer (version 2021) (2D).

## 3. Results and Discussion

### 3.1. Proximate Composition and Mineral Content of GLs

There is little information regarding agricultural production and yields of GLs in the available literature. In Korea, the average production of ginger grown in agricultural fields and domestic cultivation areas was calculated to be 14.80 ton/ha, of which GLs and other parts (excluding the root) comprised 5.76 ton/ha. Additionally, the apportionment of ginger crops by part was 39.06% roots, 15.35% leaves, and 45.59% stems. The dry yield of GLs was found to be 20.95%. Additional information regarding the nutrients and minerals found in GLs is shown in Table 1.

### 3.2. Amino Acid Contents of GLs

Generally, the quality of protein provided in a food depends on the protein content and the composition and ratio of essential amino acids. The amino acids found in GLs are listed in Table 2.

### 3.3. Flavonoid Identification Compound Content, Total Phenolic Content, Total Flavonoid Content, and RSA of GLs

Bioactive compounds extracted from GLs were analyzed qualitatively using LC/MS, and the results are expressed relative to an internal standard (6-methoxyluteolin). Nine flavonoid compounds were tentatively identified by matching their empirical molecular formulas and mass fragments (Figure 1 and Table 3). Flavonoids represent a broad group of polyphenolic compounds found in plants. These compounds have the ability to neutralize free radicals, reduce oxidative damage, and strengthen the cell’s antioxidant system. Flavonoids may also reduce the risk of DNA damage, cellular aging, and some chronic diseases. Various plant processing methods have been designed considering human health, and antioxidant substances in plant leaves have been widely characterized. Recent studies have reported the presence of antioxidants in faba bean leaves [36], mulberry leaves [37], olive leaves [38], avocado leaves [39], bamboo leaves [40], and other plants. Natural products with antioxidant activity can remove free radicals, and thus play a role in preventing cell damage caused by free radicals in the body. The binding degree of each analysis method varies depends on the nature of the product, allowing comparison and analysis of differences in antioxidant capacity [41,42]. Dose-response analyses elucidated the antioxidant potential of the extract of GLs.

### 3.4. Antioxidant-Related Effective Target Prediction

The postulated identities of the nine bioactive compounds detected were entered into the SwissTargetPrediction database to predict the protein targets. All protein targets were then entered into the UniProt database for normalization to remove repetitive targets. After deleting duplicate content, 24 final gene targets were obtained, and 4901 GeneCard and 163 OMIM targets were obtained from the online database. The gene targets were mapped to each other using the online Venny 2.1.0 software. In total, 15 (0.3%) final compound-antioxidant intersections were obtained (Figure 2). All cross-targets were located among differentially expressed genes in the antioxidant dataset.

### 3.5. Antioxidant Target PPI Network of GLs

A PPI network was built based on candidate protein targets in GLs. The PPI network in Figure 3A is the top 15 selected gene targets identified in human genes and the GO analysis. This PPI network contained 15 nodes and 62 edges. The thicker the connected line, the stronger the correlation between genes. In PPI networks, higher-level nodes may be more important for pharmacological processes. As shown in Figure 3B, the upstream human tumor necrosis factor (TNF), human prostaglandin-endoperoxide synthase-2 (PTGS2), and human interleukin-2 (IL2) were separately linked to 14, 14, and 11 proteins, respectively.

### 3.6. Build an Integrated Network

In additional analyses, an integrated visualization network was constructed containing the interactions among the bioactive compounds of GLs, antioxidant gene targets, and target-related pathways. Next, we investigated the potential mechanisms of action underlying the antioxidant effects of GLs. In Figure 4, the left rectangle represents the active compounds of GLs, the central rectangle represents genes showing antioxidant effects, and the right rectangle represents the relevant pathways of the corresponding gene targets. Connecting lines indicate that nodes can interact with each other. The network revealed potential relationships between the bioactive compounds of GLs and targets, which implied the potential pharmacological antioxidant activity of GLs. These findings suggest that a single compound affects multiple targets and that some bioactive compounds of GLs may exert their antioxidant effects through multiple targets.

### 3.7. GO and KEGG Pathway Enrichment Analyses

To verify the GO findings for the 15 highlighted targets of GLs, GO enrichment analysis of putative targets was performed and identified ‘biological processes’ (*p* < 0.001), ‘cellular composition’ (*p* < 0.05), and ‘cell composition’ (*p* < 0.05), as shown in Figure 5A,B. ‘Molecular function’ (*p* < 0.05) and ‘migration route’ (*p* < 0.05) were clarified. The y-axis represents the GO terms and the x-axis represents the number of genes enriched in those terms. Colors from blue to red represent *p* values. The size of the symbols decreases as the number of enriched genes increases in reliability and importance. We selected the top terms based on *p*-value. Enriched ‘biological processes’ included ‘negative regulation of endothelial cell proliferation’, ‘positive regulation of leukocyte adhesion to arterial endothelial cells’, ‘fever generation’, ‘response to fructose’, ‘negative regulation of the apoptotic process’, and ‘negative regulation of vascular wound healing’. ‘Cellular component enhancement included ‘cell surface’, ‘perinuclear region of cytoplasm’, ‘extracellular space’, and ‘plasma membrane’. ‘Molecular feature’ enrichment included, ’electron carrier activity’, ‘flavin adenine dinucleotide binding’, ‘oxidoreductase activity’, and ‘protein homodimerization activity’, suggesting this association. Based on their *p*-values, ‘allograft rejection’ (hsa05330), ‘C-type lectin receptor signaling’ (hsa04625), ‘human T-cell leukemia virus 1 infection’ (hsa05166), and ‘metabolic’ (hsa01100) were selected as the antioxidant pathways influenced by GLs.

### 3.8. Predicting Gene Expression of GLs Target Genes in Various Cancer Types

The effectiveness of potential antioxidant genes in plants in terms of cancer prevention can be predicted through the GEPIA (Gene Expression Profiling Interactive Analysis) website. Gene expression maps of three genes (TNF, PTGS2, and IL2) were generated for 31 sample genes that are potential targets of GL compounds (Figure 6). An upregulation of PTGS2 has been demonstrated in bladder cancer (BLCA), prostate adenocarcinoma (PRAD), lung squamous cell carcinoma (LUSC), and lung adenocarcinoma (LUAC). In general, the expression of PTGS2 is upregulated in many cancers [43,44]. The PGH2 protein product of PTGS2 is converted to PGE2 by prostaglandin E2 synthetase, which can promote cancer progression [45,46]. As a result, inhibiting PTGS2 may help prevent and treat this type of cancer. On the other hand, since TNF is responsible for tumor necrosis, upregulation of its expression in tumors by GL’s bioactive compounds might inhibit cancer cell growth. In the case of TNF, it was predicted that DLBC (diffuse large B-cell lymphoma) would be regulated at an elevated level. IL2 is a signaling receptor essential for key immune functions, such as the immune system through T-cells and the promotion of T-cell differentiation, which can help the body fight infection. These results suggest that it may be difficult to interpret the effect of a single major antioxidant compound on a single target and that individual compounds contained in plant extracts may bind to signaling receptors and simultaneously attenuate or inhibit disease factors.

### 3.9. Molecular Docking of Bioactive Compounds and Key Targets

Molecular docking analysis utilizes molecular modeling techniques to predict how proteins will interact with small molecules. In this study, molecular docking of three top-selected genes and bioactive compounds was performed. When identifying antioxidant-related genes, K3glu (the flavonoid astragalin, found in various plant parts) was selected as the representative antioxidant compound of GLs based on the probability (% and n number) of homologous genes. Astragalin has been reported to have anti-inflammatory, antioxidant, and anti-atopic dermatitis properties [47,48]. Astragalin can attenuate lipopolysaccharide (LPS)-induced inflammatory responses by inhibiting the NF-κB signaling pathway [49,50]. By convention, the binding capacity between the tested molecule and the protein in the molecular docking analysis was expressed as affinity. The affinities of the selected genes and bioactive compounds were calculated as −7.8 kcal/mol for TNF, −10.4 kcal/mol for PTGS2, and −9.5 kcal/mol for IL2 (Figure 7A–C). The docking results indicated that the receptor-ligand interactions between the candidate bioactive compounds in GLs and proteins involved hydrophobic and polar interactions: TNF in ASN-92 and SER-95; PTGS2 in SER-126, ASP-125, ILE-124, PHE-371, and SER-121; and IL2 can form hydrogen bonds in SER-126, ASP-125, ILE-124, PHE-371, and SER-121.

### 3.10. Potential Antioxidant Pathway

The C-type lectin receptor signaling pathway (Figure 8), which was judged to have high significance (*p*-value) based on the graph results in Figure 3B, was selected as the antioxidant transport pathway influenced by GLs. Flavonoid compounds detected in GLs regulate the expression of TNF and IL2 cytokines by regulating the activation of T helper 1 and 17 (Th1 and Th17) cells. This regulation plays an important role in a variety of immune and inflammatory responses. For example, quercetin is known to inhibit the expression of TNF-α and IFN-γ in Th1 cells, which may help reduce inflammation in autoimmune and inflammatory diseases [51]. Quercetin may also reduce the production of Th17-related cytokines, such as IL17 and IL22, and increase the production of Th1-related cytokines, such as IFN-γ [52,53]. We postulate that the anti-inflammatory and immunomodulatory effects of flavonoid compounds in GLs may be useful in prevention and treatment strategies for several diseases, such as chronic inflammatory diseases, autoimmune diseases, and cancer. This interpretation indicates that a single flavonoid compound in GLs exerts antioxidant effects in multiple genes that can be targeted simultaneously.

## 4. Conclusions

The DPPH and ABTS analyses indicated GLs had excellent antioxidant activity. Nine flavonoid compounds were identified in GLs through LC/MS analyses. Using a network pharmacology approach, five potential compounds and 15 relevant antioxidant targets were identified. Kaempferol was a key compound contributing to GL’s antioxidant activity. Key upstream targets were TNF, PTGS2, and IL2. ‘Allograft rejection’, ‘C-type lectin receptor signaling’, ‘human T-cell leukemia virus 1 infection’, and ‘metabolic’ were identified as important signaling pathways. These results were incorporated into a component-target-pathway network. These results may provide new insights into the mechanistic study of activities other than the antioxidant behaviors of GLs. Molecular docking showed that the potential compound astragalin has high binding energy to the target protein PTGS2. Additionally, a comprehensive method combining chemical and bioactivity assays while identifying antioxidant candidates in GLs was established. This strategy has proven to be a suitable tool for elucidating the pharmacological mechanisms of bioactive ingredients found in plants and for selecting the research and development direction of the functional material industry such as foods.

## Figures and Tables

**Figure 1 antioxidants-13-00652-f001:**
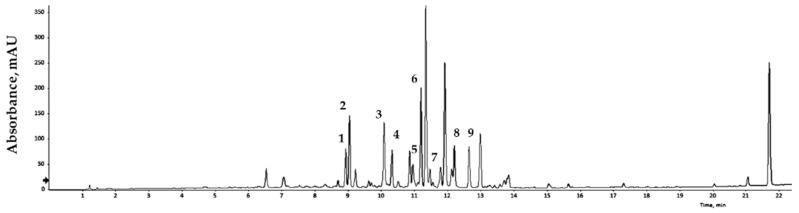
LC/MS chromatogram of the GLs.

**Figure 2 antioxidants-13-00652-f002:**
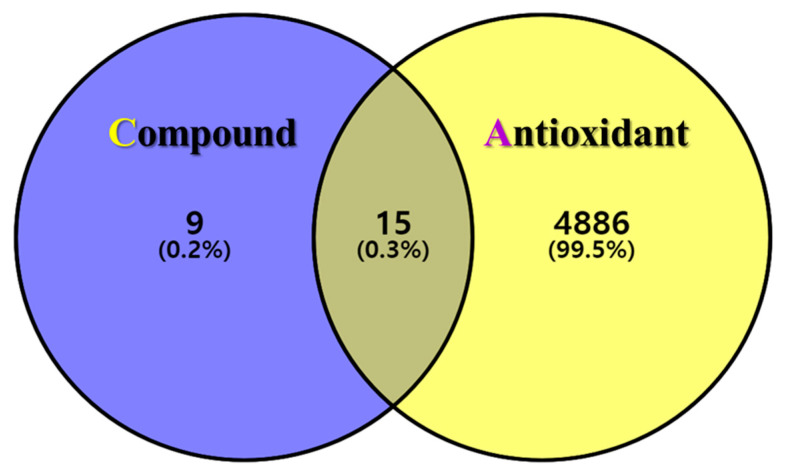
The intersection of GLs compounds with corresponding targets and antioxidant-related genes. Venn diagram of the targets corresponding to the GLs compounds (purple circle) and the antioxidant-related gene set (yellow circle). The middle is the intersection target of the two.

**Figure 3 antioxidants-13-00652-f003:**
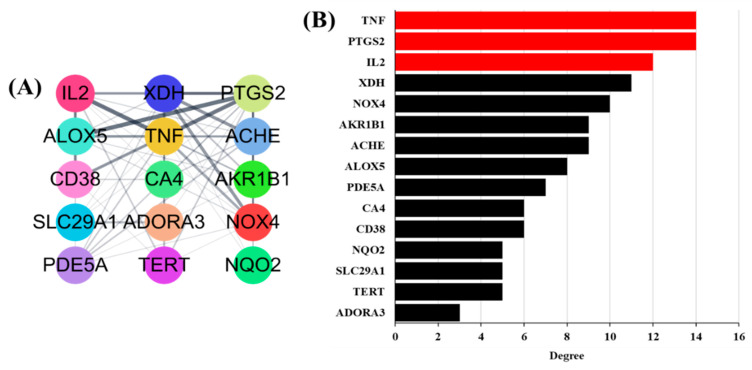
Potential antioxidant targets of PPI network: (**A**) Top 15 genes selected based on compounds. The thicker the line connecting each node, the higher the correlation. (**B**) Top 15 potential antioxidants targets ranked by degree values.

**Figure 4 antioxidants-13-00652-f004:**
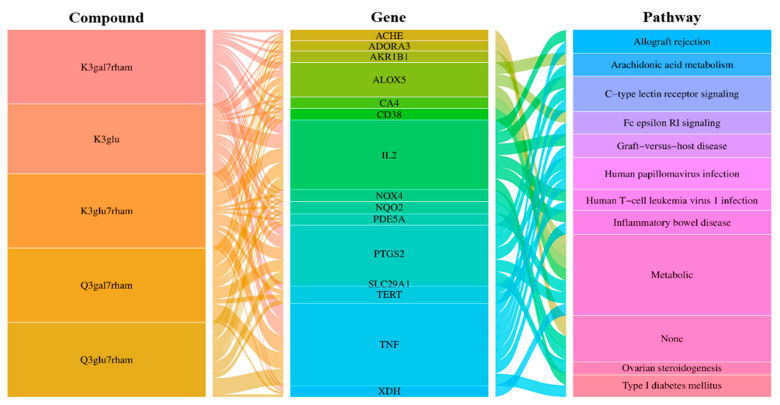
Compound-gene-pathway network of GLs for antioxidant. Interaction relationships between five selected potential compounds, 15 top genes for the compounds, and 11 signaling pathways were revealed. The left rectangles represent the bioactive compounds, middle rectangles represent the gene, and right rectangles represent signaling pathways.

**Figure 5 antioxidants-13-00652-f005:**
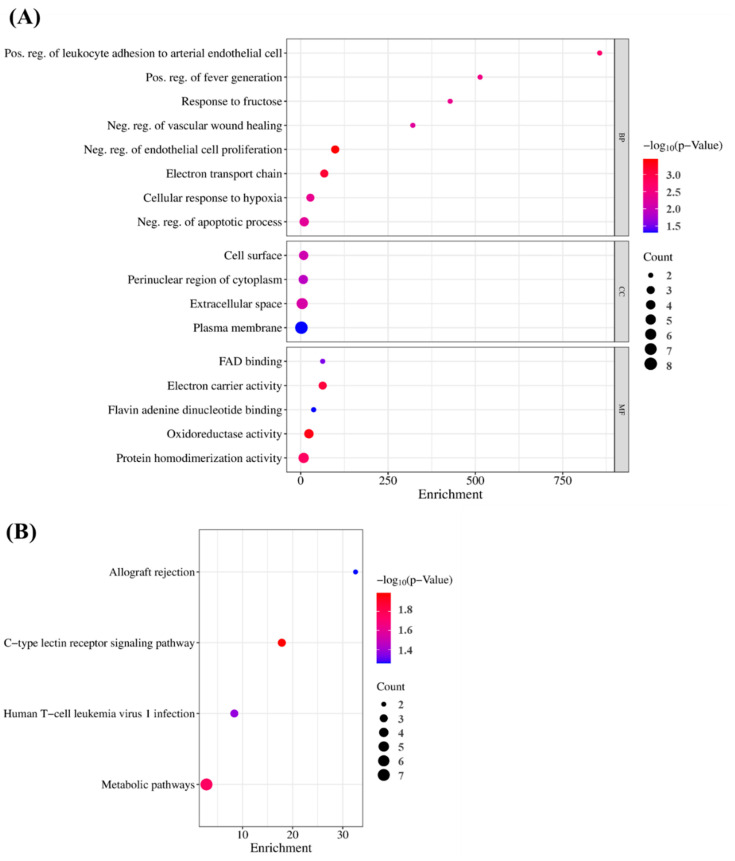
GO terms and KEGG enrichment analyses: (**A**) Gene ontology terms of 15 potential targets. The top GO functional terms were selected (BP, *p* < 0.001; CC and MF, *p* < 0.05). Abbreviations: MF, molecular function; CC, cellular component; BP, biological processes. (**B**) KEGG pathway enrichment analysis (*p* < 0.05).

**Figure 6 antioxidants-13-00652-f006:**
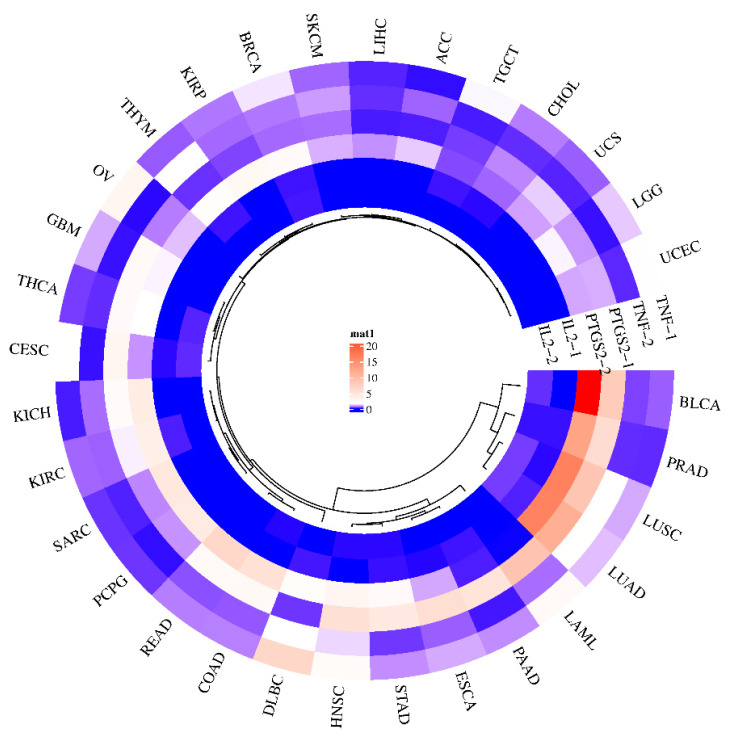
Gene expression of GLs target genes in various cancer types. Gene expression levels are shown in the center, with blue indicating low expression (minimum) and red indicating high expression (maximum). Gene-1 is indicated as a tumor factor, and gene-2 is indicated as a normal factor.

**Figure 7 antioxidants-13-00652-f007:**
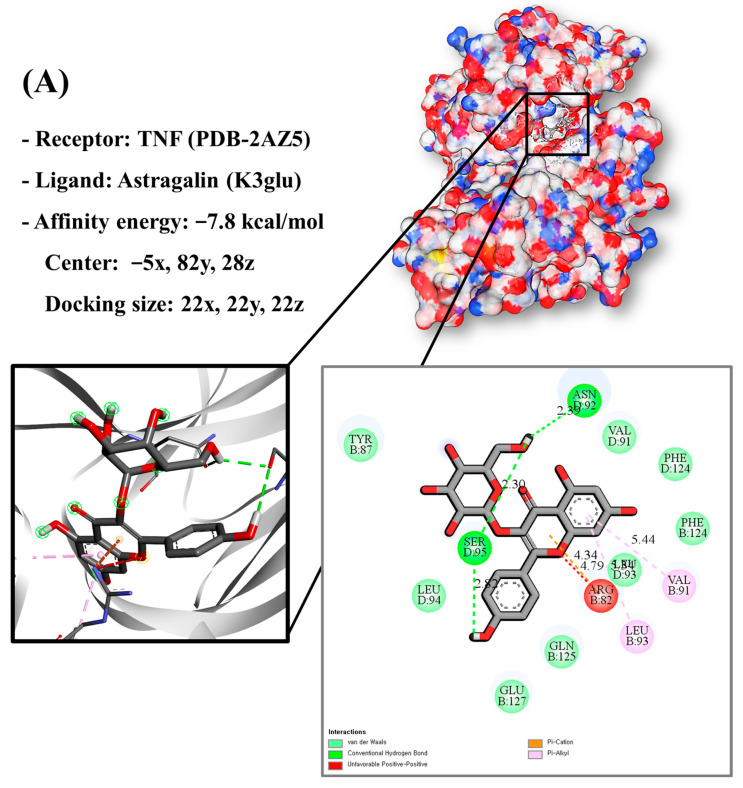
Molecular docking results of astragalin and the targets: (**A**) Docking model of TNF-astragalin with the lowest binding affinity (−7.8 kcal/mol). (**B**) Docking model of PTGS2-astragalin with the lowest binding affinity (−10.4 kcal/mol). (**C**) Docking model of IL2-astragalin with the lowest binding affinity (−9.5 kcal/mol).

**Figure 8 antioxidants-13-00652-f008:**
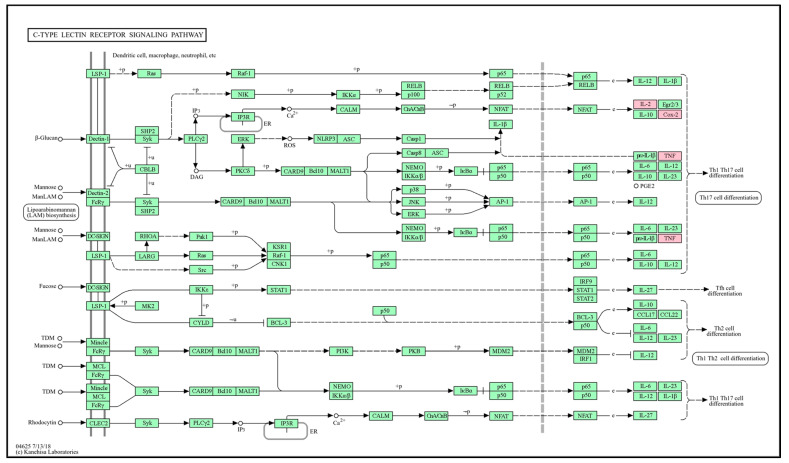
Potential antioxidant KEGG pathways of the main bioactive compounds of GLs. Simple arrows (→): Direction of a biochemical reaction. T-bar arrows (⊥): Inhibition or blocking of a reaction or pathway. Solid lines (―): Direct interactions or standard biochemical reactions. Dashed lines (− −): Indirect interactions, regulatory effects, or less certain pathways. Nodes rectangles: Genes or proteins. Those mappable within the pathway are highlighted in pink and were identified.

**Table 1 antioxidants-13-00652-t001:** Proximate composition and mineral content of GLs.

Parameters	Values
Proximate composition (%)	
Moisture	4.48 ± 0.25 ^(1)^
Crude protein	18.61 ± 0.03
Crude fat	4.51 ± 0.55
Ashes	10.05 ± 0.10
Minerals (mg/100 g)	
Calcium	866.00 ± 3.20
Copper	2.33 ± 0.03
Iron	30.40 ± 0.70
Magnesium	387.00 ± 2.30
Manganese	92.40 ± 0.60
Phosphorus	262.00 ± 1.50
Potassium	2548.00 ± 549.00
Silicic acid	2.22 ± 0.08
Sodium	33.90 ± 0.81
Zinc	3.32 ± 0.01

^(1)^ All values are mean ± SD (n = 3).

**Table 2 antioxidants-13-00652-t002:** Amino acids content of GLs.

Parameters	Values	Parameters	Values
Non-essential (mg/100 g)		Threonine	30.59 ± 0.72
Alanine	82.62 ± 0.54 ^(1)^	Tryptophan	27.09 ± 0.23
Asparagine	-	Valine	90.72 ± 0.45
Aspartic acid	29.55 ± 0.81	Conditionally essential	
Glutamic acid	112.81 ± 2.02	Arginine	1.24 ± 0.03
Serine	1.85 ± 0.23	Citrulline	11.71 ± 0.38
Essential		Glutamine	-
Histidine	12.01 ± 0.28	Glycine	23.69 ± 1.22
Isoleucine	54.31 ± 0.16	Proline	2.12 ± 0.15
Leucine	67.03 ± 0.53	Tyrosine	1.34 ± 0.08
Lysine	29.31 ± 2.34	Other amino acid	
Methionine	5.09 ± 0.34	GABA	63.12 ± 3.12
Phenylalanine	35.78 ± 0.44	Ornithine	4.09 ± 0.49

^(1)^ All values are mean ± SD (n = 3); -, not detected.

**Table 3 antioxidants-13-00652-t003:** Flavonoid compounds, total polyphenol, total flavonoid, and radical scavenging activity of GLs.

Symbol	Units, Full Name, Description	ME (ppm) ^(1)^	*m*/*z*	Values
Flavonoid identification analysis				
Q3rob7rham	%, Quercetin 3-*O*-robinobioside-7-*O*-rhamnoside, Peak 1	0.4	757.2	15.97 ± 0.05 ^(2)^
Q3rut7rham	%, Quercetin 3-*O*-rutinoside-7-*O*-rhamnoside, Peaks 2	−0.2	757.2	20.46 ± 0.08
Q3gal7rham	%, Quercetin 3-*O*-galactoside-7-*O*-rhamnoside, Peaks 5	0.7	611.2	13.02 ± 0.07
Q3glu7rham	%, Quercetin 3-*O*-glucoside-7-*O*-rhamnoside, Peaks 6	1.7	611.2	6.80 ± 0.02
K3rob7rham	%, Kaempferol 3-*O*-robinobioside-7-*O*-rhamnoside, Peaks 3	0.3	741.2	11.92 ± 0.12
K3rut7rham	%, Kaempferol 3-*O*-rutinoside-7-*O*-rhamnoside, Peaks 4	0.7	741.2	4.65 ± 0.23
K3glu	%, Kaempferol 3-*O*-glucoside, Peaks 7	2.1	449.1	0.71 ± 0.05
K3gal7rham	%, Kaempferol 3-*O*-galactoside-7-*O*-rhamnoside, Peaks 8	0.4	595.1	12.71 ± 0.04
K3glu7rham	%, Kaempferol 3-*O*-glucoside-7-*O*-rhamnoside, Peaks 9	0.8	595.1	13.77 ± 0.16
Dose-dependence test and radical scavenging activity				
TP	mg GAE/g, total polyphenol			20.44 ± 0.34
TF	mg QE/g, total flavonoid			16.46 ± 2.09
DPPH-RSA	%, DPPH-radical scavenging activity			66.93 ± 1.67
ABTS-RSA	%, ABTS-radical scavenging activity			61.90 ± 5.65

^(1)^ ME, mass error; ^(2)^ all values are mean ± SD (n = 3), flavonoid values using internal standard (6-methoxyluteolin).

## Data Availability

Data are contained within the article.

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
