# Peer review of "Health Benefits of Antioxidant Bioactive Compounds in Ginger (Zingiber officinale) Leaves by Network Pharmacology Analysis Combined with Experimental Validation"

_antioxidants, 2024, doi:10.3390/antiox13060652_

Round 1
Reviewer 1 Report
I believe that it is not necessary to repeat in the text, the values ​​mentioned in the tables
As I mentioned before, in paragraph 2.4 lines 99-100 identical to lines 85-96 in paragraph 2.3
Author Response
Thank you for your detailed review of the manuscript.
Opinion 1. I believe that it is not necessary to repeat in the text, the values ​​mentioned in the tables
Response 1. Table 1, 2, and 3 were edited.
Opinion 2. As I mentioned before, in paragraph 2.4 lines 99-100 identical to lines 85-96 in paragraph 2.3
Response 2. The section 2.3 was edited.
*Please refer to the attached file below for detailed information on the response.

Reviewer 2 Report
The paper is generally well-written and structured, based on comprehensive laboratory research. Experiments are well planned, and the analyses were affected by appropriate methods. The article is an adequate novel and interesting for publication. There is a sufficient discussion of the results obtained. All the conclusions are the logical outcome of the presented data and discussion.
As a result, I recommend that this manuscript be published in Antioxidants after minor revision (minor language editing).
The manuscript is well-written, has an important pharmacology message, and should greatly interest the readers. The results are satisfactorily presented, as well as statistical analysis. There is no need for any statistical issues with the results obtained in the manuscript.
Author Response
Thank you for your detailed review of the manuscript.
Opinion 1. I recommend that this manuscript be published in Antioxidants after minor revision (minor language editing).
Response 1. English proofreading has been completed.
*Please refer to the attached file below for detailed information on the response.

Reviewer 3 Report
The reviewed manuscript, antioxidants-3011368, presents the results of the high-quality experimental study, which is dedicated to the search for the potential of medical foods for disease treatment using the example of phytochemicals from ginger leaves. In my opinion, the paper will be interesting to readers, and I recommend it be published in Antioxidants after minor corrections.
Comments to the manuscript:
- Check the text for mistakes and typos. For example, line 35.
- Please make sure that all abbreviations are described at first mention. For example, line 84 and others.
- In my opinion, Section 2.2 is too short to be separated in alone Section. Please add more details or combine with Section 2.1.
- Please use the Equation Editor for equations 1–3.
- Please use the format defined for References on websites (lines 148, 153, 154, 158, etc.).
- Table 1 can be narrowed to the width of the main text.
- Make sure that it is necessary to duplicate the digital information in lines 199–202 and Table 1. Same for Tables 2 and 3.
- Increase the text size in the Figures. For instance, see Figs. 1 and 8. See Instructions for Authors.
- Please use “nine flavonoid” instead of “9 flavonoid”.
- Check the format of the citations in the text (see Instructions for Authors). For example, see line 34.
Author Response
Thank you for your detailed review of the manuscript.
Opinion 1. Check the text for mistakes and typos. For example, line 35.
Response 1. Edited Mistakes and typos.
Opinion 2. Please make sure that all abbreviations are described at first mention. For example, line 84 and others.
Response 2. All have been checked and corrected.
Opinion 3. In my opinion, Section 2.2 is too short to be separated in alone Section. Please add more details or combine with Section 2.1.
Response 3. Section 2.2 was deleted, combined with section 2.1.
Opinion 4. Please use the Equation Editor for equations 1–3.
Response 4. All have been checked and corrected.
Opinion 5. Please use the format defined for References on websites (lines 148, 153, 154, 158, etc.).
Response 5. All have been checked and corrected.
Opinion 6. Table 1 can be narrowed to the width of the main text.
Response 6. Table 1 is width unadjusted.
Opinion 7. Make sure that it is necessary to duplicate the digital information in lines 199–202 and Table 1. Same for Tables 2 and 3.
Response 7. Table 1, 2, and 3 were edited.
Opinion 8. Increase the text size in the Figures. For instance, see Figs. 1 and 8. See Instructions for Authors.
Response 8. Figure size was adjusted (Figure 1-8).
Opinion 9. Please use “nine flavonoid” instead of “9 flavonoid”.
Response 9. All words in this form have been corrected.
Opinion 10. Check the format of the citations in the text (see Instructions for Authors). For example, see line 34.
Response 10. All have been checked and corrected.
*Please refer to the attached file below for detailed information on the response.
